# Diversity of bacteriophages encoding Panton-Valentine leukocidin in temporally and geographically related *Staphylococcus aureus*

**Geoffrey W. Coombs**[1,2]\*, **Sarah L. Baines**[3], **Benjamin P. Howden**[4], **Krister M. Swenson**[5,6], **Frances G. O'Brien**[7,8]

**1** Antimicrobial Resistance and Infectious Diseases Research Laboratory, Murdoch University, Murdoch, Western Australia, Australia, **2** PathWest Laboratory Medicine—WA, Fiona Stanley Hospital, Murdoch, Western Australia, Australia, **3** Doherty Applied Microbial Genomics, Department of Microbiology & Immunology, The University of Melbourne at The Peter Doherty Institute for Infection & Immunity, Melbourne, Victoria, Australia, **4** Microbiological Diagnostic Unit Public Health Laboratory, Department of Microbiology & Immunology, The University of Melbourne at The Peter Doherty Institute for Infection & Immunity, Melbourne, Victoria, Australia, **5** LIRMM, CNRS-Univ. Montpellier, Montpellier, France, **6** IBC Institute de Biologie Computationnelle, Montpellier, France, **7** Curtin Health Innovation Research Institute and the School of Pharmacy & Biomedical Sciences, Curtin University, Western Australia, Australia, **8** Australian Collaborating Centre for *Enterococcus* and *Staphylococcus* Species (*ACCESS*) Typing and Research, School of Veterinary Sciences and Life Sciences, Murdoch University and Curtin University, School of Pharmacy and Biomedical Sciences, Perth, Western Australia, Australia

\* Geoffrey.Coombs@health.wa.gov.au

**Data Availability Statement:** All relevant data are within the manuscript.

## Abstract

Production of the Panton-Valentine leukocidin (PVL) by *Staphylococcus aureus* is mediated via the genes *lukS-PV* and *lukF-PV* which are carried on bacteriophage φSa2. PVL is associated with *S. aureus* strains that cause serious infections and clones of community-associated methicillin-resistant *S. aureus* (CA-MRSA) that have additionally disseminated widely. In Western Australia (WA) the original CA-MRSA were PVL negative however, between 2005 and 2008, following the introduction of eight international PVL-positive CA-MRSA, PVL-positive WA CA-MRSA were found. There was concern that PVL bacteriophages from the international clones were transferring into the local clones, therefore a comparative study of PVL-carrying φSa2 prophage genomes from historic WA PVL-positive *S. aureus* and representatives of all PVL-positive CA-MRSA isolated in WA between 2005 and 2008 was performed. The prophages were classified into two genera and three PVL bacteriophage groups and had undergone many recombination events during their evolution. Comparative analysis of mosaic regions of selected bacteriophages using the Alignments of bacteriophage genomes (Alpha) aligner revealed novel recombinations and modules. There was heterogeneity in the chromosomal integration sites, the lysogeny regulation regions, the defence and DNA processing modules, the structural and packaging modules and the *lukSF-PV* genes. One WA CA-MRSA (WA5$_{18751}$) and one international clone (Korean Clone) have probably acquired PVL-carrying φSa2 in WA, however these clones did not disseminate in the community. Genetic heterogeneity made it impossible to trace the source of the PVL prophages in the other WA clones. Against this background of PVL prophage diversity, the sequence of one group, the φSa2USA/φSa2wa-st93 group, was remarkably stable

**Funding:** This work was funded by Curtin University and the Health Department of Western Australia.

**Competing interests:** The authors declared that no competing interests exist.

over at least 20 years and associated with the highly virulent USA300 and ST93-IVa CA-MRSA lineages that have disseminated globally.

## Introduction

*Staphylococcus aureus* is a pandemic pathogen that is also part of the human microbiota [1]. Paramount to the success of *S. aureus* has been its ability to utilize mobile elements to acquire and disseminate antibiotic resistance, virulence and adaptive mechanisms amongst staphylococcal populations. In methicillin-sensitive *S. aureus* (MSSA) and community-associated methicillin-resistant *S. aureus* (CA-MRSA) the Panton-Valentine leukocidin (PVL) is a virulence factor that is carried on a bacteriophage known as ϕSa2 which is integrated into the chromosome as a prophage [2]. PVL is a bi-component, pore-forming toxin produced by co-transcribed genes, *lukF-PV* and *lukS-PV*, that targets and lyses human macrophages, polymorphonuclear leukocytes and monocytes and also incites the human inflammatory immune response [3]. Strains encoding PVL are associated with skin and soft tissue infections and dangerous invasive infections however, the role that the toxin plays in virulence is controversial and as yet, no clear-cut selective advantage has been shown for CA-MRSA that produce PVL [4–8]. Many virulent strains of MSSA and CA-MRSA do not produce PVL, however, as the pathogen evolves it is evident that those that have disseminated to cause the greatest burden of infectious disease harbor the prophage [9].

The PVL bacteriophage genome is composed of functionally colinear main modules encoding genes for lysogeny, DNA processing, head morphogenesis and packaging, tail morphogenesis, and lysis, with the *lukSF-PV* genes encoded between the lysis and lysogeny modules in the circularly permutated bacteriophage [10]. The lysogeny, *lukSF-PV* and lysis regions are well conserved with minor polymorphisms. Most diversity occurs in the DNA processing module with the head and tail morphogenesis genes showing diversity depending on the genus and PVL group. ϕSa2 can be vertically transmitted with the chromosome during replication or it can enter the lytic cycle and transmit horizontally to another cell. It has been well documented that bacteriophages undergo high rates of recombination and both these forms of transmission allow opportunity for genetic exchange, the potential mechanisms being transposition, site-specific recombination, homing endonucleases and homologous and illegitimate recombination [11]. While it is believed that horizontal gene transfer between *S. aureus* of different lineages is rare due to a lineage-specific type1 restriction-modification system [12], an *in-vivo* study revealed that bacteriophage transferred frequently during co-colonization by *S. aureus* of the same lineage and recombination between different bacteriophage occurred [13]. An investigation of MRSA colonisation in remote WA revealed that 8% of screening swab sets with an MRSA were colonised with multiple lineages of MRSA and 51.7% were co-colonised with an MSSA [14]. This would provide ideal opportunities for bacteriophage transmission and recombination to occur.

Five lineages of CA-MRSA, ST1-IVa (WA1), ST78-IVa (WA2), ST5-IVa (WA3), ST45-V (WA4) and ST8-IVa (WA5) emerged in remote Western Australian (WA) communities and WA1, WA2 and WA3 eventually disseminated to the capital city Perth and the eastern states of Australia [15, 16]. Unlike CA-MRSA that were being reported outside of WA, the WA strains were PVL negative [17]. There were however, two lineages of PVL-positive MSSA in remote WA communities, ST93-MSSA and ST121-MSSA [14].

In 2005, a PVL-positive strain belonging to the same lineage as WA1 was isolated, followed in 2008, by WA2-, WA3- and WA5-like PVL positive clones. In WA, all MRSA are submitted

to a central facility for typing and epidemiological investigation [18] and between 2005 and 2008 eight international PVL-positive CA-MRSA were introduced into WA. The rise in the number of PVL-positive CA-MRSA in WA since the first was found in 2003 has been alarming. In 2003/2004 2.1% of CA-MRSA were PVL positive, however by 2015/2016 this had risen to 52.8%, with the predominant clones being ST93-IVa (Queensland clone, 63%), ST5-IVc (WA 121, 19.5%) and ST30-IVc (WSPP, 6.8%). WA1-, WA2- and WA3-like PVL-positive clones were still in the community in 2016 however, they had not thrived and formed lower percentages of 0.7%, 0.17% and 1.1% respectively while PVL-positive WA5 had disappeared [19].

The overall aims of this study were to investigate PVL prophages from lineages of PVL-positive MRSA isolated in WA between 2005 and 2008 firstly, to gain insights into the genetics of geographically and temporally related PVL prophages in WA and secondly, to determine if PVL bacteriophages from the international strains had horizontally transmitted into the local WA clones. A comparative analysis of the PVL prophages has been performed using conventional sequence analysis, and regions of selected prophages have been compared using the Alignments of bacteriophage genomes (Alpha) aligner, which is an application that creates a partial order of gapless alignments along the bacteriophage genomes, allowing the identification of common core sequences and modular segments [20, 21]. Heterogeneity between the bacteriophages has been investigated using Alpha aligner defined modules and coding sequence comparisons. PVL bacteriophages were induced from PVL-positive *S. aureus* from the WA community and attempts were made to lysogenise prototype PVL-negative WA CA-MRSA.

## Materials and methods

### Bacterial strains

Genotypes and year of isolation of PVL-positive clones and their PVL prophage sizes are presented in Table 1. All MRSA except WA2$_{RNSH95}$ and USA300 FPR3757 were from cases of infection or colonization in the WA community [18]. WA2$_{RNSH95}$ was a WA2 clone from Sydney, Australia. The USA300 clone was present in WA [22] and the prophage φSa2USA from FPR3757 (Genbank: NC_007793) was used for genetic comparison. MSSA isolates W17S and K25S were colonizing isolates from remote WA communities [14]. ST772-V was previously sequenced [23]. MW2 (Genbank: BA000033) was used as a *lukSF-PV* gene-sequencing and prophage induction control. φSLT (Genbank: AB045978) and φSa2958 (Genbank: AP009363) were *lukSF-PV* gene sequencing controls.

PVL-negative WA1$_{WBG8287}$, WA2$_{WBG8366}$, WA3$_{WBG8378}$, WA4$_{WBG8404}$ and WA5$_{WBG7583}$ are historic prototype clones from the WA community [24]. Bacteriophage indicator and propagating strains were RN4220, WBG248, WBG356, WBG696 and WBG286.

### Sequencing of bacterial and PVL-prophage genomes and genetic analysis

Twelve bacterial genomes were sequenced using Illumina NextSeq sequence chemistry (Illumina Australia, Scoresby, Victoria 3179) and assembled with SPAdes, v3.9.0. The PVL-prophage reads were extracted and analysed using MacVector with Assembler, v15.5.3 (Accelrys, Cambridge, UK). The sequences of φSa2wa-st1, -st8, -st30, -st72 and -st93mssa were on single contigs, the remainder were assembled by overlapping contigs utilising the MacVector Assembler bowtie and phrap algorithms. Bacteriophage were designated as phi Sa2 Western Australia-host sequence type (φSa2wa-st). Except for prophages φSa2USA and φSa2wa-st772, National Centre for Biotechnology Information (NCBI) homology searches used only whole bacteriophage genome sequences for comparisons.

**Table 1. WA PVL-positive bacteriophages and lysogens.**

| Bacteriophage | Size (bp) | Lysogen genotype CC, ST-SCC*mec* | Clone$_{Strain}$ | Year of isolation | Reference |
|---|---|---|---|---|---|
| | | | WA PVL-positive Clones | | |
| φSa2wa-st1 | 45,585 | 1, ST1-IVa | WA1$_{15798}$ | 2005 | This study |
| φSa2wa-st5 | 44,823 | 5, ST5-IVa | WA3$_{18790}$ | 2008 | This study |
| φSa2wa-st8 | 45,914 | 8, ST8-IVa | WA5$_{18751}$ | 2008 | This study |
| φSa2wa-st78 | 45,878 | 88, ST78-IVa | WA2$_{RNSH95}$ | 2008 | This study |
| φSa2wa-st93mssa | 45,913 | Singleton, ST93 | W17S | 1995 | [14] |
| φSa2wa-st121mssa | 45,621 | 121, ST121 | K25S | 1995 | [14] |
| | | | International Clones | | |
| φSa2wa-st22 | 38,576 | 22, ST22-IVc | 16386 | 2007 | [18] |
| φSa2wa-st30 | 45,780 | 30, ST30-IVc | WSPP$_{16663}$ | 2002 | [25] |
| φSa2wa-st59 | 42,133 | 59, ST59-V | Taiwan clone$_{16672}$ | 2003 | [26] |
| φSa2wa-st72 | 47,213 | 72, ST72-IVa | Korean clone$_{15803}$ | 2006 | [18] |
| φSa2wa-st80 | 45,164 | 80, ST80-1Vc | European clone$_{15395}$ | 2004 | [27] |
| φSa2wa-st93 | 45,913 | ST93-IVa | Qld clone$_{16790}$ | 2003 | [28] |
| φSa2wa-st772 | 42,402 | 1, ST772-V | Bengal Bay clone$_{17048}$ | 2007 | [23] |
| φSa2USA | 45,914 | 8, ST8-IVa | USA300_FPR3757 | 2003 | [29] |
| | | | WA PVL-negative Clones | | |
| NA | NA | 1, ST1-IVa | WA1$_{WBG8287}$ | 1995 | [24] |
| NA | NA | 88, ST255-IVa | WA2$_{WBG8366}$ | 1995 | [24] |
| NA | NA | 5, ST5-IVa | WA3$_{WBG8378}$ | 1995 | [24] |
| NA | NA | 45, ST45-V | WA4$_{WBG8404}$ | 1995 | [24] |
| NA | NA | 8, ST8-IVa | WA5$_{WBG7583}$ | 1989 | [30] |

Abbreviations: bp, base pairs; NA, Not applicable; WA, Western Australian, Qld, Queensland; WSPP, Western Samoan Phage Pattern

## *lukSF-PV* sequencing

Isolates were cultured on brain heart infusion agar (BHIA) (Gibco Diagnostics, Gaithersburg, MD, USA), incubated at 37˚C, grown in trypticase soy broth (Gibco Diagnostics, Gaithersburg, MD, USA) and incubated overnight at 37˚C. DNA was extracted using the Invitrogen PureLink Genomic DNA Mini Kit (Invitrogen, Carlsbad, CA, USA) according to the instructions of the manufacturer with lysostaphin (Sigma-Aldrich, St. Louis, MO, USA) used to lyse the *S. aureus* cell wall. *lukSF-PV* was amplified as previously described [31]. Amplicons were purified using the Ultraclean DNA PCR Clean Up Kit (MoBio Laboratories, GeneWorks, Thebarton, SA, Australia) and sequences were compared with the *lukSF-PV* genes from φSLT (Genbank: AB045978).

## Bacteriophage induction and hybridisation

Bacteriophage were induced using Mitomycin C (Sigma-Aldrich, St. Louis, MO, USA) as previously described [10]. Plaques were transferred onto nylon membranes using standard techniques [32] and DNA was cross-linked to the membrane (Amersham Biosciences, Little Chalfont, Bucks, England) using a GS Gene Linker UV Chamber (Bio-Rad Laboratories, Hercules, CA, USA). Membranes were treated with 2 mg/mL Proteinase K (Roche Diagnostics, Mannheim, Germany). The hybridisation probe was obtained by PCR amplification of *lukSF-PV* using previously described primers [33]. PCR products were purified using the MoBio PCR Cleanup Kit. Probes were prepared using the DIG DNA Labelling and Detection

Kit according to the manufacturer's instructions (Boehringer Mannheim, Mannheim, Germany). Plaque hybridisation was performed as directed by the manufacturer (Boehringer Mannheim, Mannheim, Germany).

## PVL-bacteriophage propagation and lysogenisation of PVL-negative WA CA-MRSA

To propagate the mitomycin C-induced PVL-positive bacteriophages the plaques were extracted and crushed with 3 drops of BHIB, and the mixture left to stand for 10 minutes. This suspension was added to 100 μL of an overnight culture of the indicator strain, 3 mL of molten 3% BHIA was added and the mixture poured onto a BHIA plus 0.004M $Ca^{2+}$ base plate which was incubated overnight at 30˚C. The overlay containing the bacteriophage and indicator strain was scraped off and filtered. Each of the PVL-negative strains of WA CA-MRSA were grown overnight in BHIB and lawn-inoculated onto BHIA plus 0.004M $CaCl_2$. A drop of each PVL-bacteriophage lysate was placed on the lawn and incubated at 30˚C overnight. Isolated colonies growing in the centre of plaques present on lawns of PVL-negative WA clones were picked, their total DNA was isolated and lysogeny was detected using previously described primers [33].

## Results

### Sequence analysis and bacteriophage classification reveal diversity amongst the prophages

Fourteen prophage genomes between flanking direct 21 base pair (bp) repeats of 5′-AGGGCA AAAAAAGGGCg/aGATT-3′ termed *att*L and *att*R were analysed (Table 1). The 12 new prophage sequences from this study have been deposited in the NCBI database under accession numbers MF580410, MK940809 and MG029509 to MG029518.

The prophages were between 38,576 and 47,213 bp in size with between 41.4 and 100% nucleotide (nt) identity, GC compositions of 31 to 33.4% and 52 to 75 protein-coding sequences of 25 or more amino acids (aa).

The prophage genomes had the organisation of *Siphoviridae* family Sfi21-like PVL viruses of the *Caudovirales* order and, according to the most recent staphylococcal bacteriophage classification criteria, were placed into two genera and three PVL bacteriophage groups (Table 2) [34–36]. ϕSa2wa-st22, -st59 and -st772 (76.1–76.7% nt identity) were placed into the 77like-virus genus of icosahedral-headed bacteriophage. ϕSa2wa-st22 and -st772 were group 1 PVL bacteriophage with 74.4% nt identity and ϕSa2wa-st59 was group 3. ϕSa2wa-st1, -st5, -st8, -st30, -st72, -st78, -st80, -st93, -st93mssa, -st121mssa and ϕSa2USA (74.2–100% nt identity) were 3alikevirus genus, prolate-headed group 2 PVL bacteriophage. ϕSa2wa-st5 was unusual in that it encoded type C DNA polymerase (Genbank: AUM57702) rather than type A (exemplified by ϕSa2wa-st93 Genbank: AUM58245) (Fig 1).

ϕSa2wa-st93, -st93mssa (100% nt identity) and -st8 (99.97% nt identity) were considered to be the same bacteriophage as the international ϕSa2USA (99.97% nt identity), with ϕSa2wa-st72 (96.6% nt identity) very closely related. These will be known as the ϕSa2USA/ϕSa2wa-st93 group in this study. The prophages found in the WA clones, WA1$_{15798}$ (ϕSa2wa-st1), WA2$_{RNSH95}$ (ϕSa2wa-st78) and WA3$_{18790}$ (ϕSa2wa-st5) had identities of 80.8 to 93.8% and, although related, they were not identical to each other or any PVL bacteriophage in this study or in the NCBI database while ϕSa2wa-st8 from WA5$_{18751}$ had only 1 bp difference with ϕSa2USA and will be included in the ϕSa2USA/ϕSa2wa-st93 group.

**Table 2. WA PVL prophage *lukSF-PV* polymorphisms, prophage classifications and lysogen lineages.**

| Prophage | Lysogen CC, ST | Lysogen Genus/PVL gp. | SNPs | | | | | | | | |
|---|---|---|---|---|---|---|---|---|---|---|---|
| | | | *lukS-PV* | | | | | | *lukF-PV* | | |
| | | | 33 | 105 | 345 | 443 | 527 | 663 | 1186 | 1396 | 1729 |
| φSLT | 30, ST30 | 3alikevirus/2 | G | T | C | G | A | G | C | A | A |
| φSa2wa-st30 | 30, ST30 | 3alikevirus/2 | G | T | C | G | A | G | C | A | A |
| φSa2wa-st772 | 1, ST772 | 77likevirus/1 | G | T | C | G | A | G | C | A | A |
| φSa2958 | 5, ST5 | 3alikevirus/2 | G | T | C | G | A | G | C | G | A |
| φSa2wa-st1 | 1, ST1 | 3alikevirus/2 | G | T | C | G | A | G | C | G | A |
| φSa2wa-st22 | 22, ST22 | 77likevirus/1 | G | T | C | G | A | G | C | G | A |
| φSa2wa-st59 | 59, ST59 | 77likevirus/3 | G | T | C | G | A | G | C | G | A |
| φSa2wa-st8 | 8, ST8 | 3alikevirus/2 | G | T | C | G | G | T | C | A | G |
| φSa2wa-st72 | 72, ST72 | 3alikevirus/2 | G | T | C | G | G | T | C | A | G |
| φSa2wa-st93 | S, ST93 | 3alikevirus/2 | G | T | C | G | G | T | C | A | G |
| φSa2wa-st93mssa | S, ST93 | 3alikevirus/2 | G | T | C | G | G | T | C | A | G |
| φSa2USA | 8, ST8 | 3alikevirus/2 | G | T | C | G | G | T | C | A | G |
| φSa2wa-st5 | 5, ST5 | 3alikevirus/2 | G | T | C | A | A | G | C | G | A |
| φSa2wa-st78 | 88, ST78 | 3alikevirus/2 | G | C | C | G | A | G | C | G | A |
| φSa2wa-st121mssa | 121, ST121 | 3alikevirus/2 | G | T | C | G | A | G | T | A | A |
| φSa2wa-st80 | 80, ST80 | 3alikevirus/2 | A | T | T | G | A | G | C | A | A |
| φSa2mw | 1, ST1 | 3alikevirus/2 | G | T | C | G | G | T | C | A | A |

Nucleotides differing from those of φSLT are shaded. Abbreviations: gp., group

## The φSa2 chromosomal integration site was heterogenous

Chromosomal sequences proximal to the prophage terminals encoded the hybrid *att*Li and *att*Ri sites of the *att*B and *att*P sites on the chromosome and a circularly permuted form of the bacteriophage. They consist of a 29-bp central core and 25-bp left-hand (LH) and right-hand (RH) arms (Fig 2) [37]. There were nine single nucleotide polymorphism (SNP) profiles for *att*Li and seven for *att*Ri (Fig 2). Two groups of prophages shared identical *att*Li and *att*Ri sites; international prophage φSa2USA and φSa2wa-st72 with φSa2wa-st8, and Australian international prophage φSa2wa-st93 with φSa2wa-st93mssa. Of the prophage in the WA CA-MRSA-like strains, φSa2wa-st8 shared *att*Ri with the φSa2USA/φSa2wa-st93 group and φSa2wa-st772; φSa2wa-st1 had a unique *att*Li and φSa2wa-st8 shared *att*Li with φSa2wa-st72 and φSa2USA. φSa2wa-st5 and φSa2wa-st78 had unique integration-site sequences. *att*Li of φSa2wa-st78 could not be identified, however its *att*Ri was reasonably similar to the φSa2wa prophages over the LH arm and the first 17 bp of the common core (3 bp difference) while 32 of the remaining 37 bp were different (Fig 2). *att*Li of φSa2wa-st30 was absent due to a 268 bp deletion (detected by comparison with the intact "preferred integration site" of WA2$_{RNSH95}$).

With the exception of φSa2wa-st78, the bacteriophages had inserted into a gene within a cluster of three or four open reading frames (ORFs) encoding a putative domain of unknown function (DUF)1672 lipoprotein [38], one downstream of the integration site and two or three upstream. The four DUF1672 domain-containing proteins of φSa2wa-st72 had amino acid similarity scores of 61.1–79.7% indicating they were paralogues. There was variability in the truncated ORF. φSa2wa-st1, -st5, -st59, -st772 and the φSa2USA/φSa2wa-st93 group had truncated the 3' end of an ORF encoding a lipoprotein_7 superfamily domain-containing protein (54.5–100% nt identity and 55.6–100% amino acid similarity) which variably also encoded a structural maintenance of the chromosome SMC_N domain (φSa2wa-st1, -st5, -st59 and -st72). φSa2wa-

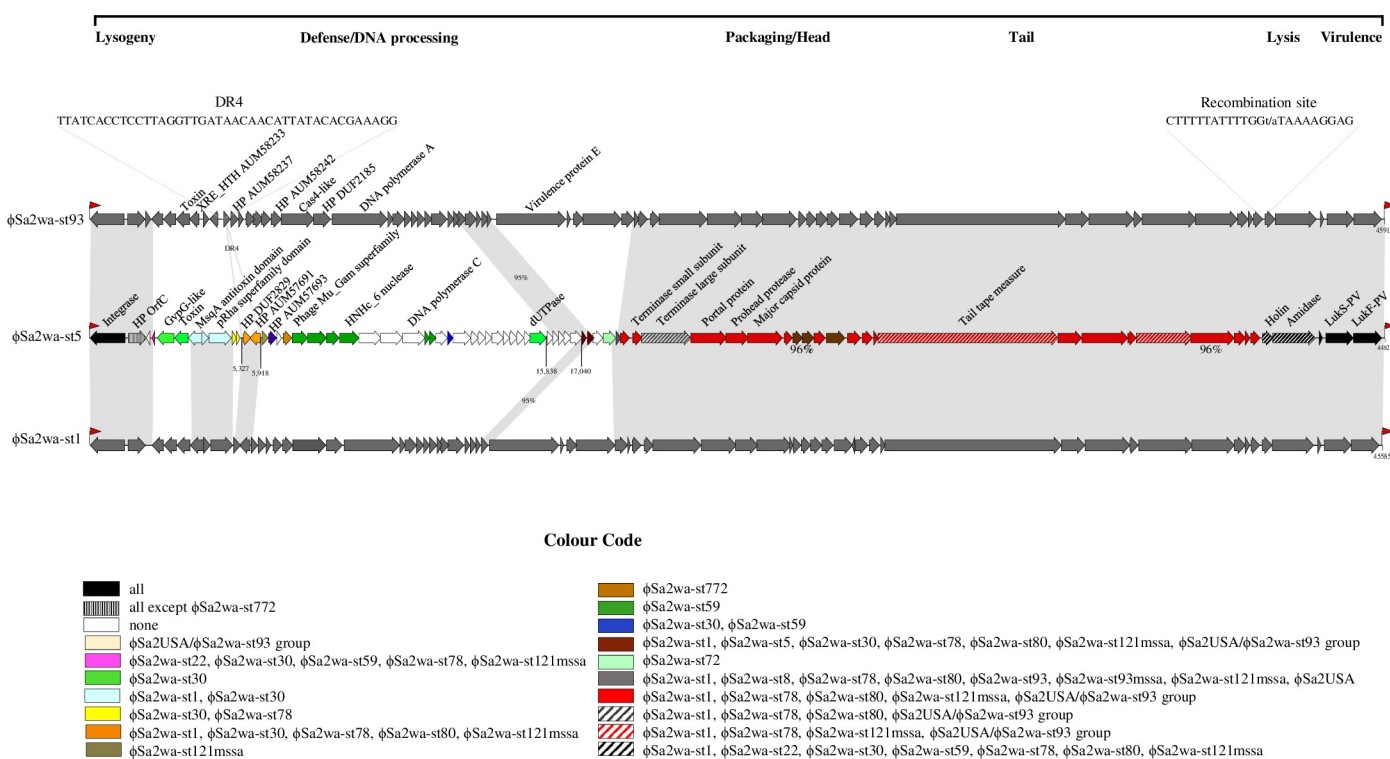

**Fig 1. Diagrammatic comparison of φSa2wa-st5 with φSa2wa-st1 and φSa2wa-st93.** φSa2wa-st5 ORFs are represented as arrows indicating the direction of transcription and coloured according to the PVL prophage or groups of PVL prophages from PVL-positive *S. aureus* in WA that share 97 to 100% nucleotide identity. Where identity is less than 97% this is indicated under the ORFs. Regions of the genomes with 97 to 100% identity with φSa2wa-st5 are shaded. Where identity is less than 97% this is indicated in the shaded region. The main functional modules are indicated on a line above the genomes. Red flags indicate *att*L and *att*R sites. Genome size is indicated at the right-hand end. Proteins encoded by ORFs relevant to this study and structural proteins are indicated. Hypothetical proteins are identified by their Genbank accession number. The positions and sequences of DR4 and a widely-shared recombination site are presented. Abbreviations: DR, direct repeat; HP, hypothetical protein.

st22, -st30, -st80 and -st121mssa, had truncated an ORF encoding a hypothetical protein (HP) which was in the same position as the lipoprotein_7 domain ORF but lacked the lipoprotein_7 domain. ORFs truncated by φSa2wa-st22, -st30 and -st80 had 82.2–94% nt identity however,

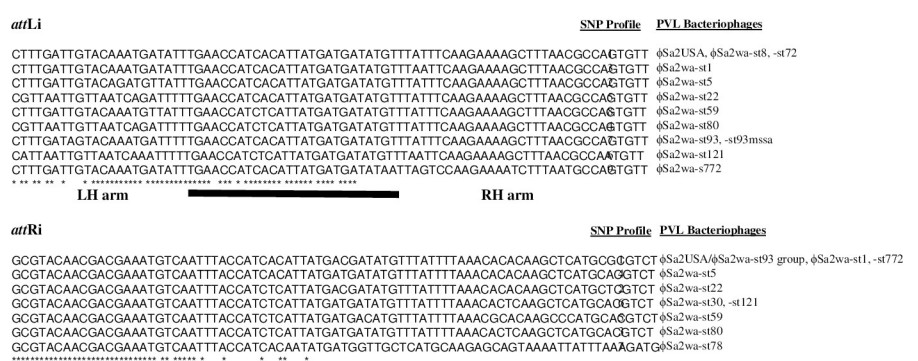

**Fig 2. Integration-site sequences proximal to the terminals of PVL prophages from PVL-positive *S. aureus* in WA (2005 to 2008).** Sequences have been aligned using ClustalW. The central core sequence is underlined with a thick black line. Left-hand and right-hand arms are indicated. SNP profiles are numbered alongside their respective bacteriophages. Identical nucleotides are indicated by an asterisk, absence of an asterisk indicates a polymorphic site. Abbreviations: LH, left hand; RH, right hand; *att*Li, left-hand integration site; *att*Ri, right-hand integration site.

the ORF truncated by φSa2wa-st121mssa had only 26.4–36.2% nt identity. Immediately upstream of all prophages except φSa2wa-st78 was an ORF encoding a 62-aa HP whose sequence indicated that it was the 3' terminal of the truncated lipoprotein_7 protein ORF, when compared with the intact lipoprotein_7 domain-encoding ORF of WA2$_{RNSH95}$ (not shown).

φSa2wa-st78 had truncated the 3' end of a 6-phospho-beta-galactosidase gene and inserted upstream of a galactose-6-phosphate degradation enzyme. The φSa2wa-st78 host genome, WA2$_{RNSH95}$, encoded an intact lipoprotein_7 domain-encoding ORF that contained an *att*Li site which was homologous over the LH arm and central core (1 bp difference) with the consensus *att*Li but had 14 bp differences in the RH arm. This may have prevented insertion of φSa2wa-st78 into what appears to be a preferred site for φSa2.

### Lysogeny regulation and modular recombination sequences

The intergenic region between the divergently transcribed integrase gene *int* and its associated HP ORF, originally called *orfC* [39] (Fig 1), contained structures indicative of involvement in regulation and lysogeny in all prophages (Fig 3). There were SNPs between the prophages, however all except φSa2wa-st772 had the same secondary structure which consisted of a consensus sigma factor H (SigH) binding-site [40] and a downstream inverted repeat (IR) of `5'-GAACGTAc/tGTTC-3'`. Overlapping the SigH binding-site was an inverted repeat that could form a possible stem-loop structure of `5'-GGGTAGgtgggCTACCC-3'` (stem-loop 1) (Fig 3). The first two nucleotides of the loop could be GT, TC or GC (φSa2wa-st772). There was then a previously identified and highly conserved stem-loop putative regulatory site, stem-loop 2 [41]. Both stem-loops were flanked by heptanucleotide direct repeats (DR) of `5'-AAAATAA-3` (DR1) the first of which comprised 7 bp of the SigH binding site.

φSa2wa-st772, which has previously been predicted to be a recombinant bacteriophage [23] had a regulation region that was somewhat different. The intergenic region was between *int* and a different HP ORF (exemplified by YP_00910342) transcribed on the same strand. The regulatory features however, included the SigH binding-site with its downstream IR and stem-loop 1; stem-loop 2 was absent and there was only one copy of DR1, which occurs from 24 to 33 times in the prophage genomes.

Of the prophages in the WA CA-MRSA the φSa2wa-st1 regulation region was identical with that of φSa2wa-st80 while φSa2wa-st5, -st8 and -st78 were identical with φSa2USA.

A previously described 23-bp recombination site that has been found in unrelated staphylococcal bacteriophage [42] was found downstream of the holin gene in all prophages (Fig 1). φSa2wa-st772 encoded the enterotoxin A gene flanked by direct repeats (DRs) of `5'-CTTTTTATTTTG-3'` immediately downstream of this site thus implicating the site in the acquisition of an extra virulence factor, probably from an unrelated family φ3 beta haemolysin-converting bacteriophage.

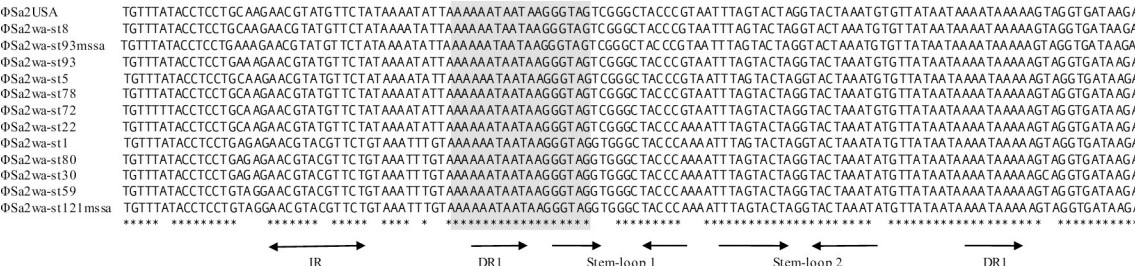

**Fig 3. Regulation regions of PVL prophages from PVL-positive *S. aureus* in WA (2005 to 2008).** Sequences have been aligned using ClustalW. The SigH binding site is shaded. Identical nucleotides in the alignment are indicated by an asterisk. Repeats are indicated by arrows. Abbreviations: IR = inverted repeat; DR = direct repeat.

ϕSa2wa-st5 has a module of 592 bp (bp 5,327–5,918) which encodes a DUF2829 protein and a HP ORF (Fig 1, Genbank: AUM57690 and AUM57691) flanked by 42-bp direct repeats (Fig 1, DR4). This module is present in ϕSa2wa-st1, -st30, -st78, -st80 and -st121mssa however, ϕSa2wa-st1, -st80 and -st121mssa lack a LH copy of DR4. Furthermore, this repeat is present as a similarly positioned single copy in all other study prophage except ϕSa2wa-st22 and -st772, indicating that the module has disseminated horizontally between bacteriophage of the same genera and the repeat is a conserved sequence that could mediate recombination, integration and excision.

## *lukS/F-PV* sequence SNPs were not specific for PVL bacteriophage genera or lysogen genotype

A single copy of a DR that has been implicated in the deletion of the *lukSF-PV* and integrase module [43] was found downstream of the *lukSF-PV* genes in all prophages however, a second copy was not be found in any of the genomes. Nine *lukSF-PV* SNPs were found, six in *lukS-PV* and three in *lukF-PV* and there were eight SNP profiles (Table 2). All except the A→G (histidine→arginine) substitution at position 527 in the ϕSa2USA/ϕSa2wa-st93 group and ϕSa2MW, were synonymous. The ϕSa2USA/ϕSa2wa-st93 group SNPs were identical. ϕSa2wa-st30 and -st772 were identical to ϕSLT. ϕSa2wa-st22, -st59 and -st1 were identical to the CC5 control ϕ2958. ϕSa2wa-st78, -st80 and -st121mssa had individual *lukSF-PV* SNPs reported previously for their respective genetic lineages [44, 45] and ϕSa2wa-st5 had a unique *lukSF-PV* SNP profile.

The distribution of the SNP profiles was heterogenous. (i) Highly similar PVL prophage with the same *lukSF-PV* SNPs lysogenised *S. aureus* of three different lineages, indicating horizontal dissemination of a successful PVL bacteriophage between *S. aureus* of three lineages; the ϕSa2USA/ϕSa2wa-st93 group lysogenised ST8-IVa, ST72-IVa, ST93, and ST93-IVa. (ii) Different PVL prophages with different *lukSF-PV* SNP profiles lysogenised *S. aureus* of the same lineage, indicating horizontal transmission of different PVL bacteriophage into *S. aureus* of the same lineage; 77likevirus, PVL group 1 (ϕSa2wa-st772) and 3alikevirus, PVL group 2 (ϕSa2wa-st1 and ϕSa2mw) prophages lysogenised CC1 strains, ST772-V and ST1-IVa respectively. (iii) Different genera of PVL prophage with the same *lukSF-PV* SNP profile lysogenised different *S. aureus* lineages, indicating that different PVL bacteriophage can carry the same *lukSF-PV* module and that either the *lukSF-PV* genes disseminate horizontally between different genera of PVL bacteriophage or random substitutions occur during replication and the fittest permutations prevail regardless of the genus of PVL prophage; 77likevirus, PVL groups 1 (ϕSa2wa-st22) and 3 (ϕSa2wa-st59) lysogenised ST22-IVc and ST59-V respectively and 3alikevirus, PVL group 2 prophage (ϕ2958 and ϕSa2wa-st1) lysogenised ST5-II and ST1-IVa respectively.

## Alpha alignment of colinear regions of ϕSa2wa-st1, -st5, -st59 and -st93 identified novel modules and heterogenous genes

The DNA processing main module is the most variable region in PVL bacteriophages and Fig 4 presents Alpha alignments of colinear sections of ϕSa2wa-st1, -st5, -st59 and -st93 from the 5' end of the bacteriophages. The region encodes conserved genes associated with lysogeny, and a variable region of early transcribed genes associated with lysogeny, bacteriophage defence and regulation. Variable genes and different colinear modules with similar functions can be detected in this graphical representation of heterogeneity.

Fig 4A has 15 alignment nodes. The anchor sequences are nodes 1 and 14 which encode *int* with the 5' end of *orfC* (exemplified by ϕSa2wa-st5, Genbank: AUM57679 and AUM57680)

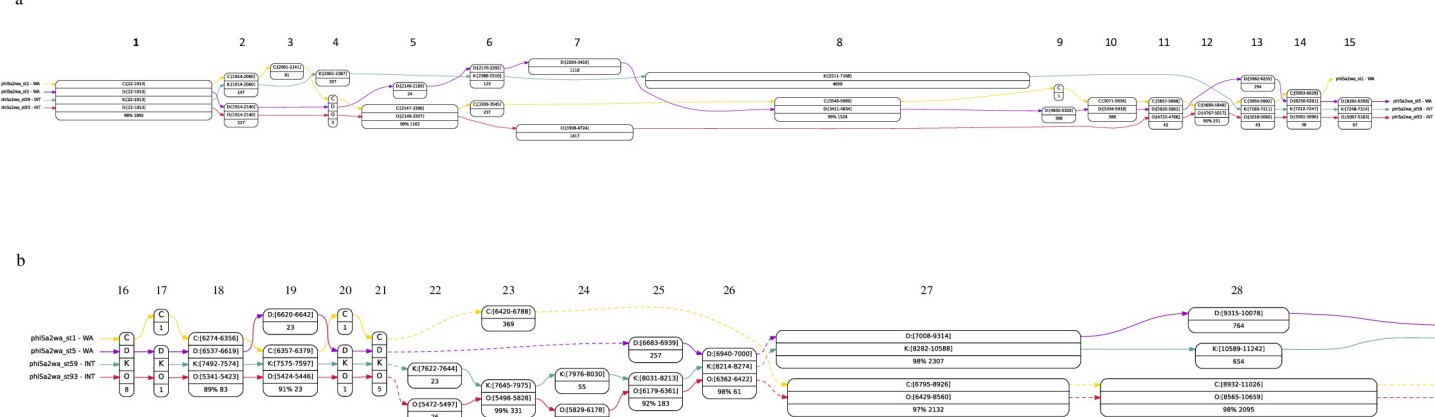

**Fig 4. A and B. Alpha alignment of colinear sections of φSa2wa-st1, φSa2wa-st5, φSa2wa-st59 and φSa2wa-st93.** Each genome in the alignment is assigned an uppercase letter. Alignment positions for the corresponding genome are indicated in parentheses alongside the letter. Anchor sequences are similar segments of significant length shared by all genomes in the alignment. Nodes are gapless alignments specific for individual genomes in the alignment; they display the length and percent identity of the aligned region; unless otherwise indicated identity is 100%. Anchors and nodes are connected by color-coded arrows, one color for each genome and numbered sequentially on the figure. Dotted arrows replace nodes of less than 20 bp. Abbreviations: C, φSa2wa-st1; D, φSa2wa-st5; K, φSa2wa-st59; O, φSa2wa-st93; WA, Western Australian; INT, International.

and the 5' terminal of a helix-turn-helix (HTH) domain-encoding ORF (Genbank: AUM576 93) respectively (Fig 1). The genomes diverge at bp 1,913 (Alpha alignment node 1). Following two or three HPs of unknown function φSa2wa-st1 (c3081-3542), -st5 (c2946-3416) and -st93 (c3080-3550) encode a putative toxin gene, (Genbank: AUM57684, 72–89% nt identity, 82.8– 94.9% aa similarity) (Fig 1). The φSa2wa-st1 and -st93 putative toxin sequences (77.3% nt identity) diverge at bp 3,308 and 3,307 respectively (Alpha alignment node 5). All polymorphisms in these ORFs are in the 5' ends and most (27/38) are non-synonymous. Non-random distribution of polymorphisms such as this indicates homologous recombination between divergent genes has probably occurred. φSa2wa-st5 has a gas vesicle protein G (GvpG) domain-encoding ORF (Genbank: AUM57683) downstream of the toxin gene (Genbank: AUM57684, Alpha alignment node 7) which has 89% nt identity with the φSa2wa-st1 toxin gene. The φSa2wa-st1 and -st5 toxin genes are followed by a xenobiotic response element (XRE)-HTH putative transcriptional regulator ORF which also encodes a MqsA antitoxin superfamily domain (Genbank: AUM57685, Alpha alignment node 8, 99.4% nt identity), while φSa2wa-st93 encodes a XRE-HTH transcriptional regulator (Genbank: AUM58233, Alpha alignment node 7). φSa2wa-st1 and -st5 then encode a XRE_HTH family regulator followed by a bacteriophage pRha superfamily domain-containing protein that interferes with infection of strains that lack integration host factor (Genbank: AUM57686 and AUM57687, Alpha alignment node 8).

When compared with all the WA-PVL prophages the node 8 module of φSa2wa-st1 and -st5 is shared with only φSa2wa-st30, and the node 7 module of φSa2wa-st93 is shared with only the φSa2USA/φSa2wa-st93 group. φSa2wa-st59 encodes a unique 4,658 bp section (bp 2511–7168), which encodes two XRE family transcriptional regulators, a bacteriophage anti-repressor and 10 putative HPs (AUM57878 to AUM57891, Alpha alignment node 8). Nodes 13, 14 and 15 encode the 5' end of a HP ORF (Genbank: AUM58237) with a common core (Alpha alignment node 14) and divergence by φSa2wa-st5 and φSa2wa-st1 indicated by the graph.

Following two to four heterogenous colinear HP ORFs the Fig 4B Alpha alignment node 27 reveals two unrelated colinear modules. φSa2wa-st5 and -st59 encode ORFs for a bacteriophage Mu Gam-like protein which protects double stranded DNA from exonuclease degradation, two

overlapping single-stranded binding proteins and a putative HNHc_6 superfamily nuclease (Genbank: AUM57696 to AUM57698) which are not shared by any other WA PVL prophage (Fig 1). φSa2wa-st1 and -st93 encode three overlapping ORFs encoding a HP, a Cas4-like protein and a DUF2185 protein (Genbank: AUM58242, AUM58243 and AUM58244). This module may be a defence system or part thereof against the bacterial CRISPR-Cas system and it is shared by all study prophage (97–100% nt identity) except φSa2wa-st5, -st22, -st59 and -st772. φSa2wa-st1 and -st93 then encode DNA polymerase A (Alpha alignment node 28) while φSa2wa-st5 (Fig 1) and -st59 encode DNA polymerase C.

## Alpha aligner-defined nodes reveal extensive mosaicism and recombination in PVL bacteriophage from WA

The singleton ST93 genome is stable and well-adapted in the geographical region. To further investigate the mosaicism in the prophages Table 3 presents the nodes of φSa2wa-st93 as determined by the Alpha aligner in Fig 4A and 4B, identifies the putative proteins or protein

**Table 3. Alpha aligner defined nodes of φSa2wa-st93 and node-associated ORFs or intergenic regions having 97–100% sequence identity with other WA PVL prophages.**

| φSa2wa-st93 Position/Node* | Protein(s) accession no's, HPs or regions | WA PVL bacteriophages with 97–100% sequence identity |
|---|---|---|
| 22-1913/1 | Integrase, AUM58227; Split HP, AUM58228 | φSa2wa-st1, -st5, -st22, -st30, -st59, -st72, -st78, -st80 -st121mssa |
| 1914-2140/2 | Split HP, AUM58228; split HP AUM58229 | φSa2wa-st5, -st72, -st78 |
| 2146-3307/5 | Split HP, AUM58229; HP, AUM58230; HP, AUM58231; split Toxin, AUM58232 | φSa2wa-st1, -st72 |
| 3308-4724/7 | Split Toxin, AUM58232; HP, AUM58233; HP, AUM58234; HP, AUM58235 | φSa2wa-st72 |
| 4725-4766/11 | Intergenic region | φSa2wa-st1, -st5, -st30, -st59, -st72, -st78, -st80, -st121mssa |
| 4767-5017/12 | HP, AUM58236 | φSa2wa-st72 |
| 5018-5060/13 | Split HP, AUM58237 | φSa2wa-st1, -st30, -st59, -st72, -st78 |
| 5061-5096/14 | Split HP, AUM58237 | φSa2wa-st1, -st5, -st30, -st59, -st72, -st78, -st80, φ -st121mssa |
| 5097-5163/15 | Split HP, AUM58237 | φSa2wa-st5, -st30, -st59, -st72, -st78, -st80, -st121mssa |
| 5341-5423/18 | Split DUF1270, AUM58238 | φSa2wa-st72, -st78 |
| 5424-5446/19 | Split DUF1270, AUM58238 | φSa2wa-st72, -st78 |
| 5472-5497/22 | Intergenic region | φSa2wa-st72 |
| 5498-5828/23 | HP, AUM58239; split DUF2482 HP, AUM58240 | φSa2wa-st59, -st72 |
| 5829-6178/24 | Split DUF2482 HP, AUM58240; split DUF1108 HP, AUM58241 | φSa2wa-st72 |
| 6179-6361/25 | Split DUF1108 HP, AUM58241 | φSa2wa-st72 |
| 6362-6422/26 | Split DUF1108 HP, AUM58241 | φSa2wa-st5, -st59, -st72 |
| 6429-8560/27 | HP, AUM58242; Cas4-like, AUM58243; DUF2815 HP, AUM58244 | φSa2wa-st1, φ -st30, -st72, -st78, -st80, -st121mssa |
| 8565-10659/28 | DNA polymerase A, AUM58245; split DUF3113 HP, AUM58246 | Sa2wa-st1, -st30, -st72, -st78, -st80, -st121mssa |

Proteins and hypothetical proteins are indicated by their Genbank protein-id number. Genbank domains of unknown function are indicated; Split proteins represent split open reading frames. Abbreviations: DUF, domain of unknown function; HP, hypothetical protein; no's, numbers

*As presented in Fig 4

**Table 4. Bacteriophage induction and lysogenisation of historic PVL-negative WA CA-MRSA.**

| Lysogen | Lysogenised recipients Total pfu/mL | | PVL positive plaques | Induced PVL bacteriophage | PVL-negative CA-MRSA lysogenised |
|---|---|---|---|---|---|
| | RN4220 | WBG286 | | | |
| MW2 | $>1x10^5$ | 0 | $>100$ | $\phi$Sa2mw | WA5$_{WBG7583}$ |
| WA1$_{15798}$ | 0 | 0 | 0 | 0 | NA |
| WA2$_{RNSH95}$ | 0 | $1x10^3$ | 0 | 0 | NA |
| WA3$_{18790}$ | $2x10^3$ | 0 | 20 | $\phi$Sa2wa-st5 | None |
| W17S | 0 | $1x10^2$ | 1 | $\phi$Sa2wa-st93mssa | Not tested |
| K25S | $3x10^2$ | 0 | 3 | $\phi$Sa2wa-st121mssa | Not tested |
| Qld Clone$_{16790}$ | 0 | $1x10^2$ | 1 | $\phi$Sa2wa-st93 | WA5$_{WBG7583}$ |

Abbreviations: NA, not applicable; pfu, plaque forming units.

sections and shows the local prophages that encode the same sequence with 97 to 100% nt identity. $\phi$Sa2wa-st8, -st93mssa and $\phi$Sa2USA are almost identical to $\phi$Sa2wa-st93 and have been excluded from the Table. There is evidence of extensive recombination. The only prophage that was identical in this region was the closely related $\phi$Sa2wa-st72 and the only prophage not to share any module was $\phi$Sa2wa-st772. Most of the shared sequence involved the prolate-headed 3alikevirus prophages however, there was also evidence of recombination with the 77likevirus icosahedral-headed $\phi$Sa2wa-st59. The Alpha aligner defined modules consist of split genes, single genes, groups of genes and intergenic regions, some shared by several prophages and others by only one or two. At 96.6% nt identity $\phi$Sa2wa-st72 is a member of the $\phi$Sa2USA/$\phi$Sa2wa-st93 group in this study and the Table 3 modules with homology only with $\phi$Sa2wa-st72 represent modules encoding functions that, amongst the prophages in this study, are unique to this successful bacteriophage.

With between 77% and 82.9% nt identity $\phi$Sa2wa-st5 from WA3 was the most distantly related of the 3alikevirus group and was successfully induced and therefore probably transmissible (Table 4). It had highest homology with $\phi$Sa2958 (Genbank: AP009363; 99% nt identity over 72% of the genome) however, this was essentially in the structural morphology and lysis, virulence and lysogeny modules. Fig 1 presents a diagrammatic comparison of $\phi$Sa2wa-st5, -st93 and -st1 with the $\phi$Sa2wa-st5 ORFs coloured according to the WA PVL-prophages that shared 97–100% sequence identity. With the exception of the bp 15,838 to 17,040 module which was homologous with non-PVL bacteriophage 53 (Genbank: AY954952; 100% nt identity) the unshared regions of $\phi$Sa2wa-st5 encoding multiple ORFs were unique. As well as random mutations that occur during chromosomal replication this heterogeneity indicates that horizontal recombination has occurred between the bacteriophage during their evolution.

### *in-vitro* induction of PVL-positive $\phi$Sa2 from Australian *S. aureus* and lysogenisation of historic PVL-negative WA CA-MRSA

To test the transmissibility of PVL-positive $\phi$Sa2 lysogenising the Australian *S. aureus* and the lysogenic capabilities of the historic PVL-negative WA CA-MRSA (Table 1), *in-vitro* induction, propagation and lysogenisation experiments were performed (Table 4).

Bacteriophage were induced from all isolates tested except WA1$_{15798}$. Overall, only two of the five indicator strains, RN4220 and WBG286 were lysogenised and specifically, only one in each induction experiment, demonstrating some specificity of lysogenisation (Table 4). Hybridisation of the plaques revealed that $\phi$Sa2wa-st5, -st93mssa, -st93, -st121mssa and the control, $\phi$Sa2mw, were induced out of their lysogens. $\phi$Sa2wa-st78 may not have been induced because

it lacked an evident *att*Li integration site (Fig 1) however, the reason why φSa2wa-st1 was not induced is currently unclear.

φSa2wa-st121mssa could not be propagated to a sufficiently high titre *in-vitro* however, φSa2wa-st5, -st93 and the control, φSa2mw, were tested for their ability to lysogenise all of the historic PVL-negative WA CA-MRSA. WA5$_{WBG7583}$ was lysogenised with φSa2wa-st93 and the control, φSa2mw, but not φSa2wa-st5. None of the other PVL-negative WA CA-MRSA were lysogenised *in-vitro* with any of the induced and propagated bacteriophages.

## Discussion

The genomes of PVL prophages from temporally and geographically related *S. aureus* of local and international origin have revealed an unexpected amount of diversity that has made it difficult to trace their origins. There has been recombination between bacteriophage of the same and different genera as well as genetic diversity in the chromosomal integration sites, the regulation regions, the defence, DNA-processing, structural and packaging modules and the *lukSF-PV* genes. There was no evidence that the icosahedral-headed prophages from international clones of CA-MRSA had transferred to the WA clones. The prolate-headed prophage formed the largest group however, with the exception of φSa2wa-st8, they were so diverse it was not possible to determine if there had been horizontal transmission of whole bacteriophages. There has been recombination between the international and local prolate-headed bacteriophage and, to a lesser extent, also between prolate- and icosahedral-headed bacteriophage that are present in WA at some stage during their evolution.

With 99.97% sequence identity, it is evident that WA5$_{18751}$ has probably acquired φSa2wa-st8 in the WA community from either a ST93 *S. aureus* or USA300. φSa2wa-st93 was induced *in-vitro* and then it lysogenised PVL-negative WA5$_{WBG7583}$ demonstrating that this clone can accept the bacteriophage. On-the-other-hand, WA5$_{18751}$ and USA300 had identical φSa2 integration-site sequences indicating that the bacteriophage could also have been horizontally transmitted from USA300.

φSa2wa-st8, -st72, -st93, -st93mssa and φSa2USA probably represent a single bacteriophage that has transmitted between lineages of *S. aureus*. USA300 and the Queensland clone are two of the most virulent and widely disseminated CA-MRSA and in this and a previous study [46] it is evident that there has been horizontal transmission of a φSa2USA/φSa2wa-st93-type bacteriophage between their CC8 and Singleton 93 ancestors, however there is no indication of when or where this occurred. USA300 acquired φSa2USA in North America following importation of its ancestor in the early 20[th] century [47]. φSa2wa-st93mssa was present in ST93-MSSA, the most prevalent colonizer in remote WA in 1995, and this clone was the ancestor of the Queensland clone that emerged in Queensland, Australia in the early 2000's [14, 48, 49]. φSa2wa-st93mssa was well adapted in ST93-MSSA and Australia before the clone acquired the SCC*mec* and before USA300 was imported into Australia [22]. Against the background of PVL prophage diversity revealed in this study it is extraordinary that the φSa2USA/φSa2wa-st93 bacteriophage has remained stable over at least 20 years in different geographic and genetic environments. To gain insights into the success of this bacteriophage it would be informative to investigate the putative proteins of unknown function encoded by the unique modules of φSa2wa-st93 revealed in Table 3.

The international Korean CA-MRSA clone is characteristically PVL-negative [50] and has probably acquired φSa2wa-st72 in the WA community. This may represent a recent acquisition of a φSa2USA/φSa2wa-st93 bacteriophage with the prophage undergoing gradual changes as it adapts to a CC72 background and different geographical conditions. As with the WA

PVL-positive CA-MRSA the PVL-positive Korean clone has not thrived and forms only 0.02% of CA-MRSA in the WA community [19].

The diversity in shared genes such as the hypothetical proteins that have been split in Table 3 according to their homologies with all the prophage in the study is interesting. In the putative toxin genes, all of the nucleotide differences were in the 5' end of the ORF and most resulted in different amino acids. This may be a defence strategy that has involved recombination within genes resulting in proteins with the same function but different antigenic profiles.

With the exception of two pairs of prophage all had distinct integration-site sequences however, the impact of this on the specificity of lysogenisation is currently unknown. The effect of lysogeny by φSa2 on host fitness could not be determined. The preferred insertion site was within a lipoprotein_7 domain-encoding ORF within a paralogous cluster of three or four ORFs that encoded a DUF1672 lipoprotein. As has been previously reported, the ORFs truncated by φSa2wa-st22, -st30 and -st80 type bacteriophage [37] and now φSa2wa-st121, were different, however they were similarly positioned within the same DUF1672 lipoprotein cluster. Lipoproteins serve as transporters of nutrients and contribute to virulence and fitness in *S. aureus* and increased complements have been associated with particularly pathogenic strains [38]. The impact of truncation of the φSa2 target genes on host fitness requires further investigation.

Prophage lysogenising 11 lineages of *S. aureus* have been investigated in this study and adaptation to different genetic backgrounds is undoubtedly one of the reasons for the diversity observed. Investigation of more genomes of PVL prophage from *S. aureus* belonging to the same genetic lineage is now required. The low occurrence of PVL-positive variants of established PVL-negative CA-MRSA in the WA community suggests that the clones may not have adapted well to the acquisition of PVL-positive φSa2.

## Acknowledgments

This work was funded by grants from the Health Department of WA and Curtin University. The authors would like to acknowledge Tam Le who performed the bacteriophage PVL gene typing and induction experiments, the scientists of the Australian Collaborating Centre for *Enterococcus* and *Staphylococcus* Species (*ACCESS*) Typing and Research for typing the isolates and provision of epidemiological information and Warren Grubb for critical reading of the manuscript.

## Author Contributions

**Conceptualization:** Geoffrey W. Coombs.

**Data curation:** Sarah L. Baines.

**Formal analysis:** Sarah L. Baines, Krister M. Swenson, Frances G. O'Brien.

**Project administration:** Geoffrey W. Coombs, Benjamin P. Howden, Frances G. O'Brien.

**Software:** Krister M. Swenson.

**Supervision:** Geoffrey W. Coombs, Benjamin P. Howden, Frances G. O'Brien.

**Writing – original draft:** Frances G. O'Brien.

**Writing – review & editing:** Geoffrey W. Coombs.

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
