## [Decision Letter · Decision Letter 0]

17 Sep 2019

PONE-D-19-19758

Diversity of Bacteriophages Encoding Panton-Valentine Leukocidin in Temporally and Geographically Related Staphylococcus aureus

PLOS ONE

Dear Dr. O'Brien,

Thank you for submitting your manuscript to PLOS ONE. After careful consideration, we feel that it has merit but does not fully meet PLOS ONE’s publication criteria as it currently stands. Therefore, we invite you to submit a revised version of the manuscript that addresses the points raised by the reviewers.

We would appreciate receiving your revised manuscript by Nov 01 2019 11:59PM. To enhance the reproducibility of your results, we recommend that if applicable you deposit your laboratory protocols in protocols.io, where a protocol can be assigned its own identifier (DOI) such that it can be cited independently in the future. For instructions see: http://journals.plos.org/plosone/s/submission-guidelines#loc-laboratory-protocols

We look forward to receiving your revised manuscript.

Kind regards,

Herminia de Lencastre, Ph.D.

Academic Editor

PLOS ONE

Journal Requirements:

Reviewers' comments:

Reviewer's Responses to Questions

**Comments to the Author**

1. Is the manuscript technically sound, and do the data support the conclusions?

Reviewer #1: Yes

Reviewer #2: Partly

2. Has the statistical analysis been performed appropriately and rigorously? 

Reviewer #1: N/A

Reviewer #2: N/A

3. Have the authors made all data underlying the findings in their manuscript fully available?

Reviewer #1: Yes

Reviewer #2: No

4. Is the manuscript presented in an intelligible fashion and written in standard English?

Reviewer #1: No

Reviewer #2: Yes

5. Review Comments to the Author

Reviewer #1: This is a highly detailed genomic analysis of recombinational interactions among PVL-encoding prophages in Western Australia (WA), including data indicating that recently imported S. aureus strains with PVL-encoding prophages contributed to the remarkable recent increase in PVL prevalence in WA. Given that the phenomenon of recombinational promiscuity among phages is very well known and has been thoroughly documented years ago, this paper represents merely a catalog of this phenomenon as applied to a particular set of prophages. Although a useful addition to the general literature on recombinational promiscuity among prophages, it does not present any novel principles or ideas. Moreover, the authors employ an elaborate prophage and strain nomenclature, which makes the presentation highly confusing, nearly impenetrable. It would help greatly if this were radically simplified. Finally, although there is a hint that the paper is about the impact of foreign strains on the prevalence and spread of PVL among prophagres in WA, it is never clearly stated as the story behind this study. If this is, in fact, the motivation, it should be clearly stated up front and detailed in the discussion.

A couple of minor points:

1. Please explain how these strains were introduced into Australia.

2. It is implied, line 236, that Sa2wa-st78 used a different att site, but the data in Fig. 1 do not support this – please clarify.

3. Were any matching sequences between otherwise unrelated genes encountered, that could account for recombination, other than that mentioned in line 276?

4. Could the authors show (tabulate) which imported PVL phages lysogenized native strains intact, and which segments of these were incorporated by recombination into native prophages?

Reviewer #2: Overview: The paper by O’Brien et al. investigates the diversity of prophages carrying PVL in different isolates of CA-MRSA from Western Australia. Before 2005, all CA-MRSA isolates from this region were PVL negative. This paper seeks to explore the genetic diversity of PVL encoding prophages from WA to try and identify their origin. The authors hypothesize that the PVL phage may have been introduced to WA strains via horizontal transmission from international PVL positive CA-MRSA strains.

The paper is excellently written and investigates, in great depth, the similarities and differences between the bacteriophages in question. There are no major concerns with the work (mostly bioinformatic) outlined in the manuscript, although some improvements could be made to improve the readability of the manuscript for non-experts in the field (comments below).

Major comments

• The major concern with this manuscript arises from the questionable impact/result of the study. The results of the study are summed up by the authors themselves in the discussion as: “The genomes of PVL bacteriophages … have revealed an unexpected amount of diversity that has made it difficult to trace their origins.” Essentially this is the concern. An in-depth analysis of 14 bacteriophages ultimately revealed there was too much diversity to make any solid conclusions regarding their origin.

• More detail. There are numerous places throughout the manuscript where insufficient information is given which makes the data difficult to read/interpret. Examples include;

o Results section “Sequence analysis”. The prophages were sorted into different genera depending on “morphogenesis and structural genes.” What were these criteria?

o There are genera in this section that are mentioned but not fully described, i.e, the “3alikevirus” genus. To me, this doesn’t mean anything without a further definition of this genus. This may be common knowledge among phage biologists but not to others.

o Titles for the results section are uninformative. Titles like “sequence analysis, PVL gene sequencing, alpha alignments etc.” do not provide readers with any information on conclusions drawn.

o The results section “Induction and lysogenisation” (and corresponding Table 4) is confusing and lacking in detail. The authors should clearly state what was done, (i.e what strains were induced), which ones they were able to obtain bacteriophage from, why they used those strains, which recipient strains they attempted to infect …etc. Several columns in Table 4 are unexplained (e.g. lukSF-PV – pos plaques. This is the only experimental section in the manuscript and it is unclear and difficult to interpret.

• Data. The authors do not indicate if the bacteriophage sequences generated have been deposited in an online repository.

Minor criticisms:

• The designation for strains/prophages used in the study is confusing and limits readability for those not familiar with this type of nomenclature. The titles are very lengthy and they make it difficult to keep track of which isolate is being discussed. While I acknowledge it may be difficult if they could be simplified it would make the paper much clearer.

• Authors alternate the spelling of “defense” and “defence” throughout the manuscript. Please adjust to make uniform.

• In vitro is not always italicized, one example is in line 421.

• Figure 4 is difficult to interpret. The colors on the key don’t always match the colors on the actual figure. Specifically, the purple color designating ϕSa2wa-st80 either doesn’t appear on the figure or is a totally different shade on the figure. The figure legend could be expanded to help explain the color code. It would also be useful to indicate the areas shown in Fig 1 and 2.

• Line 187: “Fourteen prophages…(Table 1)”. Why does Table 1 have 17 phages listed?

• Line 278: (Figure 4). Figures are out of order

• Line 262 to 275. Overall the writing in this section was unclear and I was unable to follow the impact of Sa2wa-st772

6. PLOS authors have the option to publish the peer review history of their article (what does this mean?). If published, this will include your full peer review and any attached files.

Reviewer #1: Yes: Richard. P. Novick, MD

Reviewer #2: No

---

## [Author Response · Author response to Decision Letter 0]

25 Oct 2019

We thank the reviewers for their time spent reviewing the manuscript and appreciate their suggestions. The author agrees that some parts of the presentation are confusing and difficult to read and acknowledges both reviewers for requesting simplification of nomenclature. The following changes have been made:

• In the Materials and Methods section the bacteriophage/prophage nomenclature has been explained and the author respectfully requests permission to leave it unchanged. So far there has not been any standardization of nomenclature for PVL bacteriophage in S. aureus. Following discussions with colleagues, the authors felt it was important for this and future publications to apply a meaningful nomenclature that would show at a glance which bacteriophage/prophage they were, their geographical provenance and the genetic background of their lysogen. To make the manuscript easier to read, where the bacteriophage occur in groups in the same sentence full designation has been given to only the first bacteriophage, for the remainder the prefix in front of the hyphen has been removed and only the bacteriophage -st designations separated by commas have been presented. Furthermore, to reduce the number of prophages mentioned, where the Sa2USA/Sa2wa-st93 group of bacteriophages had the same properties, the individual bacteriophages have been removed and the group designation inserted. 

• Clone designations have been shortened by presenting the strain identity number as a subscript.

• Clone genotypes have been shortened by removing “MRSA” and the bracketed SCCmec descriptors from the designations.

• In addition to the changes requested by the reviewers the author feels that the second paragraph in the PVL gene sequencing section was overly complicated and difficult to comprehend. For several years now scientists have been sequencing the PVL genes and most publications have associated different alleles with the genetic lineage of the lysogen, however, this study shows that this is not the case, therefore we feel compelled to report the findings in detail. No information has changed. The main points have been numbered and sentences have been re-arranged to make the paragraph more comprehensible.

Reviewer #1

1. This is a highly detailed genomic analysis of recombinational interactions among PVL-encoding prophages in Western Australia (WA), including data indicating that recently imported S. aureus strains with PVL-encoding prophages contributed to the remarkable recent increase in PVL prevalence in WA.

The study was performed due to concern that the PVL bacteriophages from international MRSA clones were moving into the predominant local clones. We did not mean to imply that the international clones were responsible for the high rate of PVL-positive MRSA in WA. To address this issue, we have modified the text as follows:

• In the Abstract, a sentence has been added. “There was concern that PVL bacteriophages from the international clones were transferring into the local clones, therefore a comparative study of PVL-carrying Sa2 prophage genomes from historic WA PVL-positive S. aureus and representatives of all PVL-positive CA-MRSA isolated in WA between 2005 and 2008 was performed.”

• In the Introduction: Sentence change. “The rise in the number of PVL-positive CA-MRSA in WA since the first was found in 2003 has been alarming.”

2. Given that the phenomenon of recombinational promiscuity among phages is very well known and has been thoroughly documented years ago, this paper represents merely a catalog of this phenomenon as applied to a particular set of prophages.

• What this study is trying to do is further the scientific understanding of bacteriophage recombination and heterogeneity by using the Alpha aligner to identify modules and investigate at the sequence level exactly where the greatest diversity is, what form it takes and which modules are being exchanged.

• We acknowledge the known high rate of recombination by bacteriophages and in the Introduction extra information has been added as follows: 

“It has been well documented that bacteriophages undergo high rates of recombination and both these forms of transmission allow opportunity for genetic exchange, the potential mechanisms being transposition, site-specific recombination, homing endonucleases and homologous and illegitimate recombination (11).”

Minor points

1. Please explain how these strains were introduced into Australia.

• In the Introduction: New sentence. “In WA, all MRSA are submitted to a central facility for typing and epidemiological investigation (21) and between 2005 and 2008 eight international PVL-positive CA-MRSA were introduced into WA.”

2. It is implied, line 236, thatSa2wa-st78 used a different att site, but the data in Fig. 1 do not support this – please clarify.

• There is no attLi for Sa2wa-st78 in Fig 2. It could not be identified. 

• We have commented on this in the following unchanged sentence: 

 “ attLi of Sa2wa-st78 could not be identified, however its attRi was reasonably similar (3 bp difference) to the Sa2wa prophages over the LH arm and the first 17 bp of the common core while 32 of the remaining 37 bp were different (Fig 2)”

• In the last paragraph of this section the difference is further discussed.

3. Were any matching sequences between otherwise unrelated genes encountered, that could account for recombination, other than that mentioned in line 276?

• There were matching sequences throughout however, given the plethora of mechanisms of recombination employed by bacteriophages we would not dare to speculate on how any recombinations may have occurred unless we could show evidence of the event (such as flanking repeats).

4. Could the authors show (tabulate) which imported PVL phages lysogenized native strains intact, and which segments of these were incorporated by recombination into native prophages?

• A Table would be too small. Only one PVL-positive native WA clone had a bacteriophage that was identical (97-100% homology) to one of the imported bacteriophages. That was WA518751 lysogenised with Sa2wa-st8 which has been incorporated into the Sa2USA/Sa2wa-st93 group.

• Given the heterogeneity of the bacteriophages, the vastness of the geographical region and community they came from, and how very little is understood about bacteriophage recombination it would not be possible from this study to reliably predict where individual segments originated.

Reviewer #2

Major comments

1. The major concern with this manuscript arises from the questionable impact/result of the study. The results of the study are summed up by the authors themselves in the discussion as: “The genomes of PVL bacteriophages … have revealed an unexpected amount of diversity that has made it difficult to trace their origins.” Essentially this is the concern. An in-depth analysis of 14 bacteriophages ultimately revealed there was too much diversity to make any solid conclusions regarding their origin.

• With the exception of the Sa2USA/Sa2wa-st93 group the authors were unable to get any bootstrap support for any phylogenetic analysis on this collection of PVL bacteriophage. This is remarkable in itself. The authors feel that the PVL prophage may be providing the lysogens with fitness properties other enhanced virulence from PVL production. This is why this study has looked so closely at the chromosomal attachment region and searched for novel or unique bacteriophage genes and modules. This will be for future studies.

2. The prophages were sorted into different genera depending on “morphogenesis and structural genes.” What were these criteria? There are genera in this section that are mentioned but not fully described, i.e, the “3alikevirus” genus. To me, this doesn’t mean anything without a further definition of this genus.

• The most recent reference on staphylococcal bacteriophage classification has been used. The classifications are based on the criteria therein. To direct readers to the references more clearly we have re-written a sentence as follows:

 “The prophage genomes had the organisation of Siphoviridae family Sfi21-like PVL viruses of the Caudovirales order and, according to the most recent staphylococcal bacteriophage classification criteria, were placed into two genera and three PVL bacteriophage groups (Table 2) (34-36).”

• The most striking difference between the genera is that the 77likevirus genus consists of icosahedral-headed bacteriophage and the 3alikevirus genus are prolate-headed bacteriophage. This has been mentioned in the text and reinforced throughout the document.

3. Titles like “sequence analysis, PVL gene sequencing, alpha alignments etc.” do not provide readers with any information on conclusions drawn.

• The Results headings have been changed as requested.

4. The results section “Induction and lysogenisation” (and corresponding Table 4) is confusing and lacking in detail. The authors should clearly state what was done, (i.e what strains were induced), which ones they were able to obtain bacteriophage from, why they used those strains, which recipient strains they attempted to infect …etc.

• This section has been re-written to address the reviewers’ concerns. To keep consistency with the main focus of the study only induction experiments on the Australian clones have been reported and more details on what was done have been included. The author agrees that presenting the results of the international clones only added confusion. The Table has been re-formatted to include only the Australian clones with the control, and an extra column has been added to present the results of the different recipient/indicator strains.

• One sentence giving extra information has been added:

“The induced bacteriophages each lysogenised only one of the two indicator strains demonstrating some specificity of lysogenisation.”

5. Data. The authors do not indicate if the bacteriophage sequences generated have been deposited in an online repository.

• In the original submission, this data was under the heading “Sequence data availability” at the end of the Results section. It has now been moved to the first paragraph of the Results section to make it more prominent.

Minor criticisms

1. Authors alternate the spelling of “defense” and “defence” throughout the manuscript. Please adjust to make uniform.

• This has been done.

2. In vitro is not always italicized, one example is in line 421.

• This has been done.

3. Figure 4 is difficult to interpret. The colors on the key don’t always match the colors on the actual figure. Specifically, the purple color designating ϕSa2wa-st80 either doesn’t appear on the figure or is a totally different shade on the figure. 

• The colors have been adjusted. 

4. The figure legend could be expanded to help explain the color code.

• The following sentence has been adjusted in the Figure 4 legend:

“Sa2wa-st5 ORFs are represented as arrows indicating the direction of transcription and coloured according to the PVL prophage or groups of PVL prophages from PVL-positive S. aureus in WA that share 97 to 100% nucleotide identity.”

5. It would also be useful to indicate the areas shown in Fig 1 and 2.

• The Figure 1 (now Fig 2) sequences are attachment sites that are on the chromosome, proximal to the terminals of the prophages. This is explained in the text. “Chromosomal sequences proximal to the prophage terminals encoded the hybrid attLi and attRi sites of the attB and attP sites on the chromosome and a circularly permuted form of the bacteriophage.” Because the prophage sequences were extracted from whole genomes it was not possible to determine which hybrid att site belonged to the bacteriophage and which was on the chromosome.

• The Figure 2 (Now Fig 3) region is in a part of Figure 1 that already is “cluttered” with information. The relevant ORFs are mentioned in the text and marked on Fig 1. To guide readers to the region in the text we have inserted a reference to Fig 1 as follows:

 “The intergenic region between the divergently transcribed integrase gene, int, and its associated HP ORF, originally called orfC (39) (Fig 1), contained structures indicative of involvement in regulation and lysogeny in all prophages (Fig 3).”

6. Fourteen prophages…(Table 1)”. Why does Table 1 have 17 phages listed?

• Table 1 in the manuscript submitted has 14 prophages listed. The resubmitted manuscript has Table 1 with 14 prophages.

7. Line 278: (Figure 4). Figures are out of order

• The Figures have been re-numbered and re-inserted in the correct order

8. Line 262 to 275. Overall the writing in this section was unclear and I was unable to follow the impact of Sa2wa-st772

• This is the first description of the secondary genetic features which are predicted to be involved in lysogeny and regulation in Sa2 bacteriophages. The authors respectfully feel that the text, although somewhat complicated is supported by the figure and provides a guide for further investigations.

• The impact of the differences in Sa2wa-st772 have been addressed in a new paragraph in this section.

---

## [Decision Letter · Decision Letter 1]

11 Dec 2019

PONE-D-19-19758R1

Diversity of Bacteriophages Encoding Panton-Valentine Leukocidin in Temporally and Geographically Related Staphylococcus aureus

PLOS ONE

Dear Dr. O'Brien,

Thank you for submitting your manuscript to PLOS ONE. After careful consideration, we feel that it has merit but does not fully meet PLOS ONE’s publication criteria as it currently stands. Therefore, we invite you to submit a revised version of the manuscript that addresses the points raised during the review process by reviwer #2

We would appreciate receiving your revised manuscript by Jan 25 2020 11:59PM. To enhance the reproducibility of your results, we recommend that if applicable you deposit your laboratory protocols in protocols.io, where a protocol can be assigned its own identifier (DOI) such that it can be cited independently in the future. For instructions see: http://journals.plos.org/plosone/s/submission-guidelines#loc-laboratory-protocols

We look forward to receiving your revised manuscript.

Kind regards,

Herminia de Lencastre, Ph.D.

Academic Editor

PLOS ONE

Reviewers' comments:

Reviewer's Responses to Questions

**Comments to the Author**

1. If the authors have adequately addressed your comments raised in a previous round of review and you feel that this manuscript is now acceptable for publication, you may indicate that here to bypass the “Comments to the Author” section, enter your conflict of interest statement in the “Confidential to Editor” section, and submit your "Accept" recommendation.

Reviewer #2: (No Response)

2. Is the manuscript technically sound, and do the data support the conclusions?

Reviewer #2: Yes

3. Has the statistical analysis been performed appropriately and rigorously? 

Reviewer #2: Yes

4. Have the authors made all data underlying the findings in their manuscript fully available?

Reviewer #2: Yes

5. Is the manuscript presented in an intelligible fashion and written in standard English?

Reviewer #2: Yes

6. Review Comments to the Author

Reviewer #2: My major concern with the previous version of the manuscript has not been addressed. While the authors present an extremely detailed analysis of 14 bacteriophage genomes there doesn't appear to be any solid conclusion or impact from the study. The manuscript has been modified to make it clearer and easier to understand but the overall impact of the study remains unclear.

7. PLOS authors have the option to publish the peer review history of their article (what does this mean?). If published, this will include your full peer review and any attached files.

Reviewer #2: No

---

## [Author Response · Author response to Decision Letter 1]

20 Jan 2020

Reviewer #2: My major concern with the previous version of the manuscript has not been addressed. While the authors present an extremely detailed analysis of 14 bacteriophage genomes there doesn't appear to be any solid conclusion or impact from the study. The manuscript has been modified to make it clearer and easier to understand but the overall impact of the study remains unclear.

PVL-positive CA-MRSA have evolved to become one of the most significant bacterial pathogens in global communities. From an initial concern that well adapted local clones of PVL-negative CA-MRSA were acquiring the PVL-carrying bacteriophage from imported international CA-MRSA and consequently, the implications for morbidity and therapeutic management of infections in WA, the authors have told a story of the PVL-carrying bacteriophage in a defined geographical region from a point in time when there were no PVL-positive CA-MRSA to a situation where 52.8% of all CA-MRSA in WA were PVL positive. So far this is the only study that has addressed the epidemiology and genetics of a temporally and geographically related virulence-carrying staphylococcal bacteriophage.

Heterogeneity amongst bacteriophages is acknowledged and for the majority of bacteriophages in this study this is what was ultimately revealed however, one bacteriophage was shown to be remarkably stable over at least 20 years in different genetic backgrounds. In a time of the emergence of multiple antibiotic resistance in S. aureus, when several groups are investigating the utility of bacteriophages as therapeutic agents, this finding is important. If bacteriophages are to be used as therapeutic agents then their epidemiology and basic biology, such as regulation of lysogeny, need to be understood. Why is this bacteriophage so stable and successful in particularly virulent CA-MRSA? This study has used the Alpha Aligner to identify unique genes that may help answer this question. Furthermore, use of the Alpha Aligner has added to the understanding of basic bacteriophage biology by enabling a detailed analysis of the genetics of the most diverse region of the bacteriophages.

There is a paucity of information on the molecular biology of staphylococcal bacteriophages and the author respectively feels that the overall impact of the manuscript is that it that it presents new knowledge that can be used for further investigations into bacteriophage biology and genetics in CA-MRSA.

Changes to the manuscript

• The Abstract has been modified. The following statement has been removed because the authors did not feel it was relevant to the abstract:

“…however the role played by the toxin and the bacteriophage in S. aureus virulence has been controversial.”

• In the abstract the clones that probably acquired the PVL bacteriophage in the WA community have been named and reference to the fact that they did not disseminate has been included.

• To enable clearer assessment of the impact of the manuscript the focus of the study has been made clearer in the Introduction by numbering and a more logical reversing of the two aims. This is followed through in the Abstract.

• Throughout the manuscript reference to the chromosomal attachment site as the “attachment site” has been changed to “integration site”. This is to avoid confusion with the cell wall bacteriophage attachment site that is used in many publications.

• A comma has been deleted and another inserted to improve grammar and readability of the following sentence. “Overall, only two of the five indicator strains, RN4220 and WBG286 were lysogenised and specifically, only one in each induction experiment, demonstrating some specificity of lysogenisation (Table 4).”

• In the Discussion. In the first sentence the word “prophages” has replaced “bacteriophages”.

• In reference 44. The Journal has been correctly abbreviated.

---

## [Editor Report · Decision Letter 2]

22 Jan 2020

Diversity of Bacteriophages Encoding Panton-Valentine Leukocidin in Temporally and Geographically Related Staphylococcus aureus

PONE-D-19-19758R2

Dear Dr. Frances O´Brien,

We are pleased to inform you that your manuscript has been judged scientifically suitable for publication and will be formally accepted for publication once it complies with all outstanding technical requirements.

With kind regards,

Herminia de Lencastre, Ph.D.

Academic Editor

PLOS ONE
---

## [Editor Report · Acceptance letter]

29 Jan 2020

PONE-D-19-19758R2 

Diversity of Bacteriophages Encoding Panton-Valentine Leukocidin in Temporally and Geographically Related Staphylococcus aureus 

Dear Dr. O´Brien:

I am pleased to inform you that your manuscript has been deemed suitable for publication in PLOS ONE. Congratulations! Your manuscript is now with our production department. 

With kind regards,

on behalf of

Dr. Herminia de Lencastre 

Academic Editor

PLOS ONE